# Spectral Analysis Approach for Assessing Accuracy of a Low-Cost Air Quality Sensor Network Data

Vijay Kumar[1], Dinushani Senarathna [1], Supraja Gurajala [2], William Olsen [3], Shantanu Sur [4], Sumona Mondal [1], and Suresh Dhaniyala *[5]

[1]Department of Mathematics, Clarkson University, Potsdam, NY, 13699, USA
[2]Department of Computer Science, State University of New York, Potsdam, NY, 13676, USA
[3]Department of Civil and Environmental Engineering, Clarkson University, Potsdam, NY, 13699, USA
[4]Department of Biology, Clarkson University, Potsdam, NY, 13699, USA
[5]Department of Mechanical and Aeronautical Engineering, Clarkson University, Potsdam, NY, 13699, USA

**Correspondence:** *Suresh Dhaniyala(sdhaniya@clarkson.edu)

**Abstract.**

Extensive monitoring of $PM_{2.5}$ is critical for understanding changes in local air quality due to policy measures. With the emergence of low-cost air quality sensor networks, high spatio-temporal measurements of air quality are now possible. However, the sensitivity, noise, and accuracy of field data from such networks are not fully understood. In this study, we use spectral analysis of a two-year data record of $PM_{2.5}$ from both the EPA and Purple Air (PA), a low-cost sensor network, to identify the contribution of individual periodic sources to local air quality in Chicago. We find that sources with time periods of 4, 8, 12, and 24 hours have significant but varying relative contributions to the data for both networks. Further analysis reveals that the 8- and 12 hour sources are traffic-related and photochemistry-driven, respectively, and that the contribution of both these sources is significantly lower in the PA data than in the EPA data. The presence of distinct peaks in the power spectrum analysis highlights recurring patterns in the air quality data; however, the underlying factors contributing to these peaks require further investigation and validation. We also use a correction model that accounts for the contribution of relative humidity and temperature, and we observe that the PA temporal components can be made to match those of the EPA over the medium- and long-term but not over the short-term. Thus, standard approaches to improve the accuracy of low-cost sensor network data will not result in unbiased measurements. The strong source dependence of low-cost sensor network measurements demands exceptional care in the analysis of ambient data from these networks, particularly when used to evaluate and drive air quality policies.

## 1 Introduction

Air pollution is one of the world's leading risk factors for disease and premature death. An estimated 16% of total global deaths in 2015 can be attributed to diseases caused by air pollution(Landrigan et al., 2018). Of particular concern is the mass concentration of Particulate Matter (PM) smaller than 2.5 $\mu m$, i.e., $PM_{2.5}$, or fine particles. Exposure to $PM_{2.5}$ has been directly correlated to diseases such as respiratory diseases and even mortality (Li et al., 2018; Xing et al., 2016; Samoli et al.,

2005; Ostro et al., 2006; Lewis et al., 2005). The high health impact of $PM_{2.5}$ is because of their ability to penetrate deep into the lungs and because their composition is often carcinogenic (Li et al., 2014). The European Study of Cohorts for Air Pollution Effects (ESCAPE) shows that exposure to high $PM_{2.5}$ concentrations is linked with a risk of developing lung cancer
(Raaschou-Nielsen et al., 2013). In addition to chronic diseases, exposure to $PM_{2.5}$ might also impact our response to acute diseases such as COVID-19 (Wu et al., 2020; Zhou et al., 2021; Mondal et al., 2022; Chaipitakporn et al., 2022). Accurate knowledge of $PM_{2.5}$ exposure and efforts to mitigate it are critical to protecting public health.

In the United States, the Environmental Protection Agency (EPA) monitors air quality by measuring regulated or criteria pol-
30 lutants including ambient $PM_{2.5}$ concentrations using Air Quality Monitoring Stations (AQMSs). The $PM_{2.5}$ measurements are made using a range of instruments classified as federal reference methods (FRMs) or federal equivalent methods (FEMs) (Noble et al., 2001). FRM refers to the specific monitoring methods that have been designated by the EPA as the reference standard for measuring air pollutants, while FEM refers to alternative monitoring methods that have been deemed equivalent to the FRM methods by the EPA. The two methods may utilize different instruments or measurement techniques but have
35 demonstrated comparability in accuracy and reliability. The strict maintenance and calibration routines followed in these stations ensure high-quality data and comparability between different locations (Castell et al., 2017). Even in the US, with over 5000 AQMSs, the geographic coverage of these monitoring sites is inadequate. The siting of AQMS is often biased towards populated areas, disadvantaging smaller cities and underdeveloped regions (Ardon-Dryer et al., 2020). Even in populated areas, the limited number of sites do not capture the high spatial variation in $PM_{2.5}$ concentrations that are likely, resulting in an
40 incorrect estimate of exposure and resultant health effects (Wang et al., 2015).

For accurate exposure assessment, an air quality monitoring network providing measurements at high spatio-temporal resolution is required. To address this need, researchers, communities, organizations, and individuals have been deploying low-cost air quality sensors that provide air quality data at a granular level not possible with the EPA AQMSs (Commodore et al., 2017;
Woodall et al., 2017). One of these networks is composed of sensors from Purple Air (PA). The PA sensing platform incorporates a pair of Plantower PMS 5003 low-cost sensors, which use laser light scattering techniques to determine ambient aerosol concentrations. The PMS5003 reports a variety of particle concentration metrics including $PM_1$, $PM_{2.5}$, and $PM_{10}$ (Sayahi et al., 2019; Ouimette et al., 2022; He et al., 2020). The PA provides two $PM_{2.5}$ values - one labeled as: cf_1 (higher correction factor) or cf_atm (atmosphere). The two values have different "correction factors" that convert the sensor light scattering mea-
surements to PM. For RH less than 70%, both values yield similar results for $PM_{2.5}$ less than 25 $\mu g/m^3$. Outside this range, when the cf_atm and cf_1 start to disagree (Barkjohn et al., 2021). It's important to note that the specific algorithm employed by PA for converting Plantower data into mass concentration, whether using cf_1 or cf_atm correction factors, has not been publicly disclosed (Ouimette et al., 2021). PA sensors also have two channels, namely A and B, that measure the exact same PM measurements. These two channels allow for the robustness of data collection by minimizing any data noise, loss of data
due to sensor failure, or measurement error due to sensor electronics issues (PurpleAir, 2020). While the low-cost sensors have

the advantage of deployment ease, their accuracy and precision are variable (Kuula et al., 2017).

The PA provides two PM$_{2.5}$ values - one labeled as: cf_1 (higher correction factor) or cf_atm (atmosphere). The two values have different "correction factors" that convert the sensor light scattering measurements to PM. For RH less than 70%, both values yield similar results for PM$_{2.5}$ less than 25 $\mu g/m^3$. Outside this range, when the cf_atm and cf_1 start to disagree (Barkjohn et al., 2021). It's important to note that the specific algorithm employed by PA for converting Plantower data into mass concentration, whether using cf_1 or cf_atm correction factors, has not been publicly disclosed (Ouimette et al., 2021).

The various PM sensors used in low-cost monitors are all subject to biases and calibration dependencies, with some factors accounted for with moderate success (e.g. meteorology, age of sensor) and others poorly (e.g. aerosol source, composition, refractive index) (Giordano et al., 2021). The PA sensor measurements are often calibrated/corrected by co-location with a reference monitor at a regulatory site (Wallace et al., 2021; Stavroulas et al., 2020; Kelly et al., 2017). Additionally, researchers have developed correction models to account for the impact of environmental conditions on sensor performance (Barkjohn et al., 2021; Ardon-Dryer et al., 2020). The deployment of PA sensors has resulted in expanding the availability of PM$_{2.5}$ data and enabling a range of studies, including, validation of high resolution, large-scale regional modeling efforts (Bi et al., 2020) and understanding of the impact of wildfire smoke on local and regional air quality (Gupta et al., 2018).

Co-locating low-cost sensors with reference monitors provides a fast way for their calibration. Typically, this is done by co-locating the sensors for a period of time and then determining a scaling factor or equation based on a regression analysis. The time period for co-location is generally chosen to be around days to weeks and this allows for the calibration to be independent of data noise. The selection of the calibration time period can, however, bias the sensor data to be most sensitive to sources primarily responsible for pollutant concentration variability in that time period. Sources with shorter time periods, relative to the calibration period, are averaged out and inadequately accounted for in the calibration. Thus longer time scale events are completely lost in the calibration process.

Published studies on low-cost sensors have observed some of the above mentioned problems. The response characteristics of low-cost sensors are seen to be different from that advertised by their manufacturers, possibly because the aerosol size distributions and compositions differ with location (Kuula et al., 2020; Tryner et al., 2020). As an example, low-cost sensor data are seen to be in better agreement with reference monitors at locations with low traffic than those at high-traffic locations (Castell et al., 2017). To improve the quality of the reported data from low-cost sensor networks, we need to establish ideal field calibration principles for these units. For this, frequency based methods that have been previously used in air quality to find prominent temporal components can be used (Hies et al., 2000; Marr and Harley, 2002; Choi et al., 2008; Tchepel and Borrego, 2010). Time-series decomposition using low-pass filters can identify pollution sources that account for most of the measurement variation (Zhang et al., 2018; Bai et al., 2022). Here, using frequency-based analysis, the dependence of low-cost sensor PM$_{2.5}$ measurement accuracy on the calibration period will be established.

For this work, we chose our study area as Cook County, IL, which includes the City of Chicago and has a total population of nearly 10 million. Cook County is a major transportation hub lying at the crossroads of the country's rail, road, and air traffic, and an important industrial center, thus, there are a number of emission sources within the area. Despite a baseline long-term trend of improving air quality in Chicago, recent years show a worsening trend. $PM_{2.5}$ concentrations have nearly doubled since 2017, rising from 6.7 $\mu g/m^3$ in 2017 to 12.8 $\mu g/m^3$ in 2019, exceeding the U.S. EPA air quality standards (12 $\mu g/m^3$) (IQAIR, 2020). The likely reason for increase in $PM_{2.5}$ levels is the associated increase in emissions from mobile sources in recent years (Milando et al., 2016). The changing air pollution levels have increased public interest in air quality monitoring, particularly using low-cost sensor networks. For the time period starting May 2018, the PurpleAir network in Chicago and its surrounding neighborhoods has increased from a few sensors to more than 30 sensors now.

In this study, we used $PM_{2.5}$ data from EPA sites and PA sensors located in Cook County, IL to understand differences in their data as a function of sensor location and time. Using spectral theory, we extract temporal signatures of the EPA and PA data and analyze their differences as a function of time period to determine the effectiveness and limitations of the current approach to correct low-cost sensor data to match EPA data. The results of this analysis will help us understand biases in the data from low-cost sensors such as PA networks and provide guidance in devising new approaches to field calibrate data from these sensors.

## 2   Materials and Methods

### 2.1   Data Collection and Pre-processing

Cook County, IL, has 14 EPA air quality monitoring sites, providing data on criteria pollutants, including ambient $PM_{2.5}$ concentrations (EPA, 2021). Hourly $PM_{2.5}$ measurements from EPA are available at 7 out of 14 monitoring sites in Cook County, IL. The PA network in Cook County consists of more than 30 PA low-cost sensors, that currently provide $PM_{2.5}$ data (PA, 2021). Our analysis was conducted using data from a time period of October 2019 to September 2021. For this time period, hourly $PM_{2.5}$ data was only available at 10 out of 30 PA sensors. Further, after eliminating sites with more than 20% missing data, our analysis could only use data from 5 EPA sites and 9 PA sensors, as shown in (Figure 1) and in (Table S1)

It was observed that PA data included some outliers with very large $PM_{2.5}$ concentrations, which are likely erroneous data. To eliminate these outliers from our analysis, we chose a data range of [0,70] $ug/m^3$ as valid data (Ardon-Dryer et al., 2020). In (Figure 1a) the sampling locations of EPA and PA are plotted on the map with the population density around the sampling locations in (Figure 1b). The population density in census blocks, as defined by U.S. Census Bureau (Bureau, 2021), was calculated using ArcgisPro 2.8. From a simple analysis of the siting of sensors, it is clear that the more than 60 % of PA sensors are located in urban areas where the average population density is more than 5000, exceeding that of the EPA sites except for EPA site E2.

## 2.2    Standard Correction Model

It has been established that low-cost sensors are sensitive to meteorological parameters, especially relative humidity (Barkjohn et al., 2021; Ardon-Dryer et al., 2020). This is because PA measurements are based on light scattering, with factory calibration to convert measurements to $PM_{2.5}$ values. As the composition and size distribution of particles in Chicago is likely different from that used in the sensor calibration, the reported values will need some correction. Additionally, temperature and relative humidity can alter particle physical and optical properties that PA measurements are sensitive to. While EPA measurements will also be affected by these air properties, the impact is lower because of thermal and humidity conditioning of samples prior to measurements. (Zheng et al., 2018; Kelly et al., 2017; Magi et al., 2020). Recently a US-wide correction model for PA sensors that takes into account the contribution of ambient conditions on sensor performance was introduced (Barkjohn et al., 2021). The model was built using data from 53 PA sensors, with data spanning the time period of September 2017 to January 2020, at 39 distinct sites spread throughout 16 states. From an evaluation of several models using temperature and relative humidity, they suggested a final model only considering the effect of relative humidity (RH) on PA sensor data. This model, herewith called the standard correction model, is:

$$PM_{2.5} \text{ Std\_Corr} = 0.524 \, \text{PA}_{cf\_1} \, PM_{2.5} - 0.0862 \, \text{RH} + 5.75, \tag{1}$$

where, cf_1 is the higher correction factor, and RH is the relative humidity in percent. In our study, the corrections were made to the PA data using the relative humidity (RH) reported by the 9 PA sensors themselves.

## 2.3    Monitoring Data Summary

This study uses 2 years of $PM_{2.5}$ data from 5 EPA sites and 9 PA sensors from October 2019 to October 2021. Sample time series trend in $PM_{2.5}$ from a set of EPA and PA sites (EPA site E2 and PA sensor P6) that are in close vicinity (within 2 km) to each other is shown in (Figure 2a). The gap in the total time series of $PM_{2.5}$ data around April 2020 in E2 and September-October, 2021 in P6 is due to missing observations in the time series in (Figure 2a). The major causes for missing air pollutant data in reference monitor includes monitor malfunctions and errors, power outages, computer system crashes, pollutant levels lower than detection limits, and filter changes (Imtiaz and Shah, 2008; Hirabayashi and Kroll, 2017). For low-cost sensors, approximately 40 % of the data generated is missing, most likely because of extreme weather events, battery failure, and disruption in internet accessibility at sensors location (Kim et al., 2021; Rivera-Muñoz et al., 2021).

The data from both networks show high temporal variations along with some seasonal trends over longer timescales. A direct comparison of the two data sets (Figure 2b) for the combination of E2 and P6 sites shows that on average the raw PA data overestimates the EPA data by roughly 40%, consistent with previous findings. Use of the standard correction results in a decrease in the reported PA values. The resultant best-fit linear model suggests that the corrected data slightly underestimates the actual $PM_{2.5}$ by roughly 30 %.

The overall distribution of $PM_{2.5}$ data at each of the EPA and PA sites over the entire time period of our analysis is shown in (Figure 3,Table S2). The median values of $PM_{2.5}$ reported by the PA sites are always higher and more variable than that from EPA sites in the region. The median $PM_{2.5}$ values from the average from the 5 EPA sites in the region is 8.4 $ug/m^3$ while the PA data reports a median value of 10 $ug/m^3$. With the standard correction it is seen that the variability is reduced and the median is 6.9 $ug/m^3$, 20% lower than the EPA value.

While the accuracy of the correction model can be improved with some local tuning, it is clear the model did not improve the quality of fit. This suggests that the correction model does not account for all of the causes of discrepancy between the two data sets. In particular, a regression based model will not be able to account for the sensitivity of the sensors to particle compositions and hence to different emission sources. A preliminary validation of model dependence on composition can be obtained from the evaluation of model performance for the prediction of $PM_{2.5}$ concentrations during weekdays and weekends. The differing strengths of some emission sources between weekdays and weekends are expected to result in slightly different aerosol populations during these two time periods. Here, we separated the data as weekday and weekend and applied the correction model to get corrected PA data for each of the data sets. A two-sample t-test between the EPA and corrected PA data (Figure 4) shows a statistically-significant difference between the two data sets (p-value $< 0.05$ and the exact p-value = 0.000007) on weekdays but not on weekends (p-value = 0.13), providing some initial validation that the correction model does not account equally for the contribution of all sources.

To better understand the causes of model under-performance and to determine the primary drivers for this discrepancy, a frequency-based analysis is helpful. Such an analysis can help extract the contribution of any periodic emission sources that might exist and establish if the standard correction model provides a bias-free correction for all of these components.

## 3 Spectral Analysis

In meteorology and air quality studies, spectral analysis has been used to extract and examine different temporal components in the obtained data (Hies et al., 2000; Marr and Harley, 2002; Choi et al., 2008; Tchepel and Borrego, 2010). Here, using spectral analysis we determine the effectiveness of the correction model to improve the correlation of PA data with EPA data over the entire range of emission sources that contribute to Cook County's $PM_{2.5}$ population.

To ensure stationarity of the time series data, i.e., their statistical properties such as mean, variance, and autocovariance remain constant, we use the augmented Dickey-Fuller (ADF) test method (Wang et al., 2021; Lian and Ma, 2013).

The discrete Fourier transform, $X(k)$, of hourly time series $X_t$, can be calculated using the Fast Fourier transform (FFT) algorithm. The power spectral density (PSD) for a finite time series can then be calculated as the squared magnitude of $X(k)$:

$$\Phi(v_k) = |X(k)|^2 = \left| \frac{1}{\sqrt{N}} \sum_{t=0}^{N-1} X_t e^{-2\pi i v_k t} \right|^2 \tag{2}$$

where $k = 0, 1, ..., (N-1)$. $N$ is the number of observations and $v_k = \frac{k}{N}$.

For a measurement resolution of 1 hour, a wave with a period of 2 hours or more is required (Nyquist theorem). For spectral analysis using FFT, successive equal length sequences are required without any missing observations (Dilmaghani, 2007). Here

we replace the missing data points from the EPA and PA data sets using the ARIMA model with Kalman filter (Hadeed et al., 2020; Afrifa-Yamoah et al., 2020; Wijesekara and Liyanage, 2020; Saputra et al., 2021). The PSD of each EPA and PA hourly time series of $PM_{2.5}$ data was then calculated using the stats package in R.

## 3.1   Spectral Analysis: Results and Discussion

We determined the power spectral density (PSD) of $PM_{2.5}$ data for three data sets - EPA, PA, and corrected PA data - for
all of the locations available. Then, the average PSDs for each of the data sets were determined by averaging the individual PSDs of the different locations in each network. By averaging over the different locations, the PSDs in (Figure 5) represent the power spectrum of air quality over the entire Cook County area. The PSD shows that for both networks (EPA and PA), power is higher in long time periods than in short-time periods. Thus, the predominant variation in $PM_{2.5}$ data reported by both networks over the studied duration is driven by their long-term trend. The PA data is seen to have lower power compared to
the EPA at smaller time periods. Calculating the root mean squared error (RMSE) between the EPA PSD values and the two PA data sets, it is seen that the PSD of the corrected PA data has a 58% lower RMSE than the uncorrected PA. Thus, applying the US-wide EPA correction model (Equation (1)) to the PA data reduces the PA PSD error relative to the EPA over the entire range of frequencies.

    At small time periods, both networks show distinct peaks at 4, 8, 12, and 24 hours, as seen in (Figure 6). These peaks likely
represent the contribution of periodic aerosol sources, such as traffic and photochemistry, and diurnal weather patterns to the local air quality. For ease of direct comparison, we removed the baseline trend in each of the datasets, details about baseline removal are provided in Section A of supplementary document. The PSD peak heights at the 4 time periods are observed to be higher for the EPA data than the PA standard corrected data. The PSD peaks at the 4 specific time periods were then obtained for each of the 5 different EPA sites and 9 different PA sites, and are shown in (Figure 7a). The EPA data peaks are seen to be
consistently higher than the PA corrected data for all 4 time periods (4, 8, 12, and 24 hours) and higher than the PA raw data for all time periods except 12 hours. For the different time periods, the ratio of the median of PSD peaks of the 5 EPA sites to the corresponding values for the 9 PA sites is shown in (Figure 7b). For the raw PA data, the PSD values for the 4 time periods range, relative to the corresponding EPA values, range from 0.66 to 2.5. After correction, the PA peaks are seen to reduce to below 0.4 for all time periods, suggesting that the correction model suppresses these peaks.

We speculate that the 4-hour and 8-hour peaks corresponds to traffic sources and the 12 hour peak represents the contribution of secondary aerosols formed due to photochemistry and possible diurnal changes in winds and humidity (Jia et al., 2017; Hollaway et al., 2019; Tchepel and Borrego, 2010). The 8-hour peak in the raw data is seen to be similar to the EPA data, but the correction results in reducing the peak substantially. The 12 hour peak is highly over-represented in the raw data, but the correction model, like for the 8-hour peak, decreases the 12-hour contribution. The mean sizes of particles formed due to
photochemistry are likely larger than the traffic aerosol, resulting in their relatively higher efficiency of detection in low-cost PM sensors (He et al., 2020). The over-correction of the 12 hour peak that results in its significant suppression, suggests that these particles are likely less hygroscopic than the average particles. The 24 hour peak likely represents harmonics of the 8 hour and 12 hour signals, and hence represents a combination of both sources.

To confirm that the 8 hour peak is traffic and the 12 hour peak is likely to be driven by photochemistry, we analyzed changes in these peaks for weekend/weekday and winter/summer. The EPA weekday data was considered as Monday 12am to Friday 11:59 pm and weekends as Saturday 12 am to Sunday 11:59 pm. The Winter data was generated as Dec/Jan/Feb and Summer as Jun/Jul/Aug. The PSD peaks for the two time periods were then calculated and relative changes are shown in (Figure 8). The weekend 8-hour PSD peak is seen to be nearly only 60% lower than on weekdays, consistent with changes in traffic patterns expected between the two time periods (Blanchard et al., 2008) and confirming that this peak is indeed traffic related. Seasonally, the 8-hour peak does not change significantly, again largely consistent with the expectation that traffic patterns are not overly dependent on seasons. The 12 hour peak also changes weekends vs weekdays but has a greater change seasonally than that observed with the 8-hour peak. The seasonal change points to the likely contribution of photochemistry to the 12 hour peak, but the slight change of this peak between weekends and weekdays also points to contribution from other sources, including possible traffic. In addition, the 4 and 6 hour peaks are also likely related to traffic patterns (Sun, 2014).

From the 8-hour PSD peak ratios, it can be concluded that the corrected PA data is significantly underrepresented in traffic-related particles, with PSD value for the corrected PA data being only around 17% of that of the EPA PSD value for this time period (Figure 7b). This finding is consistent with general observations in previous studies that low-cost sensor measurements more closely match reference monitors at locations with low traffic than at high-traffic locations (Castell et al., 2017).

## 4  Local Correction Model

Some of the imperfections of the correction model could be attributed to the fact that the model was based on data from a wide range of locations with different emission characteristics and meteorology. Consequently, it could be hypothesized that a local correction model tuned to local conditions will result in a better correction of PA data. Additionally, as the standard correction model is built based on daily data, it could also be hypothesized that the sub-24 hour components may not be well accounted for. To determine if the sub-24 hour components in the PA data could be better matched with EPA data, we built an hourly local correction model using the same approach used in building the standard correction model (Barkjohn et al., 2021). The model was built using PA data from various selected locations and data from the nearest EPA site, with relative humidity and temperature included as predictors. Typically in MLR models, we would only consider independent variables and it could be argued that temperature and relative humidity are not entirely independent. But from a particulate matter perspective, the differing impact of these parameters make them independent of each other. Relative humidity directly affects particle size and hence measurements by low-cost sensors, such as PA. Temperature, however, has a more complex connection to particle properties. Temperature directly affects particle size and composition by modulating condensation/evaporation, which can affect PM measurements by both EPA and low-cost sensors. Temperature also indirectly affects PM properties at a location through its relation to local meteorology, especially wind direction, and hence the distribution of sources at the measurement location. To establish the independence of these parameters, we calculated the Variance Inflation Factors (VIFs) for temperature and relative humidity and these were found to be below 5. These small VIF values indicate a low level of multicollinearity for the two parameters(Ros-Gálvez, 2017) and permit their inclusion in the MLR model. A stepwise forward selection algorithm

was used to build multiple linear regression (MLR) models. A 10-fold cross-validation technique was employed by repeating the process a total of 5 times. This method of cross-validation involves dividing the data into 10 equally sized folds, and training the model on 9 of the folds while using the remaining fold as a hold-out test set. This process is repeated 10 times, with each fold serving as the test set once. By repeating the process 5 times, the robustness of the developed model is increased by training and testing it on different subsets of the data.

The obtained equation for the local correction model is:

$$PM_{2.5}\ \text{Loc\_Corr} = 0.44\ \text{PA}_{cf\_1}\ PM_{2.5} - 0.026\ \text{RH} + 0.023\ \text{temperature} + 19.76 \tag{3}$$

where $PA_{cf\_1}$ represents the PA data with the higher correction factor cf_1 reported at a specific sensor, and RH and temperature are obtained from the PA network.

After obtaining the model, its performance was evaluated using several metrics: $R^2$, root mean square error (RMSE), and mean absolute error (MAE) (see supplementary material for details about these metrics). The model performances of the standard correction and local correction models are summarized in (Table S3). The effectiveness of the local correction model in improving the accuracy of the PA data and addressing the problem of under-accounting of high frequency sources such as traffic must be ascertained.

## 5   Time Series Decomposition

For a full model evaluation, its performance will be determined for three time period components: less than 12 hours (short-term), 12 hours to a month (medium-term), and more than a month (long-term). The short-term component represents the changes in PM$_{2.5}$ data due to high frequency sources such as traffic and short-term weather events. The medium-term component accounts for variations within time periods between 12 hours and a month. The long-term component primarily captures low frequency emissions such as those related to seasonal changes in weather and meteorology, and changes in emission rates over time. (Rao and Zurbenko, 1994; Rao et al., 1997; Wise and Comrie, 2005).

To separate the time series data into the 3 components of short-term, medium-term, and long-term time periods, we use the Kolmogorov–Zurbenko (KZ) filter technique (Rao and Zurbenko, 1994), as was done in several recent PM$_{2.5}$ studies(Bai et al., 2022; Fang et al., 2022; Zhang et al., 2018; Sá et al., 2015). The KZ filter is a low-pass filter produced through repeated iterations of moving average with parameters moving window (m), and iterations (p) also known as $KZ_{m,p}$:

$$Y_t = \frac{1}{m} \sum_{j=-k}^{k} X_{t+j} \tag{4}$$

where $Y_t$ is a filtered time sequence; $X_t$ is the input time series; $k$ is the number of values included on each side of the targeted value, $m = 2k + 1$ is window length; $t$ is the time index, and $j$ is the time point of sliding.

The output of the first pass then becomes the input for the next pass. Adjusting the window length and the number of iterations makes it possible to control the filtering of different scales of motion (Eskridge et al., 1997; Milanchus et al., 1998). To filter

a period of fewer than N days, the following criterion is applied to determine the filter's effective width (Wise and Comrie, 2005):

$$m \times p^{1/2} \leq N \tag{5}$$

Also, the filter can be used to remove frequencies below a desired cutoff frequency $w_0$ (Rao et al., 1997):

$$w_0 \approx \frac{\sqrt{6}}{\pi} \sqrt{\frac{1-(1/2)^{1/2p}}{m^2-(1/2)^{1/2p}}} \tag{6}$$

The cutoff period can be obtained by $\frac{1}{w_0}$. For our study, we have used the following equations to get long-term, medium-term, and short-term components of the time series of PM$_{2.5}$ data as defined by (Hogrefe et al., 2000; Kang et al., 2008)

The long-term PM$_{2.5}$ (PM$_{2.5,B}$) component is obtained as:

$$PM_{2.5,B}(t) = KZ_{900,5}PM_{2.5}(t) \tag{7}$$

The medium-term PM$_{2.5}$ (PM$_{2.5,M}$) component is obtained as:

$$PM_{2.5,M}(t) = KZ_{3,3}PM_{2.5}(t) - KZ_{13,5}PM_{2.5}(t) \tag{8}$$

The short-term PM$_{2.5}$ (PM$_{2.5,S}$) component is obtained as:

$$PM_{2.5,S}(t) = PM_{2.5}(t) - KZ_{3,3}PM_{2.5}(t) \tag{9}$$

## 5.1 Time Series Decomposition: Results and Discussion

We separated the time series of PM$_{2.5}$ data from EPA, PA, and standard and local corrected PA data (Equations (7) to (9)) into the three time periods of long-term, medium-term, and short-term in (Figure 9). A comparison of the long-term component signals shows that the two-year trends of the PA raw data is different from that of the EPA data (Figure 9a). The correction models both lower the mean of the PA data. The standard correction is, however, seen to over-correct for mean, and does not capture the signal density accurately. Using the local model results in largely replicating the long-term PM$_{2.5}$ distribution, except at the lowest values. This might suggest that long-term changes might be driven by more than humidity, and including the effect of temperature on sensor performance could be important. In addition to air properties, long-term changes may also be driven by drift in sensor performance, which could be captured with a local model but not a standard model. In the medium-term, the standard correction model shifts the mean PM$_{2.5}$ values, in contrast to the long-term component, to align reasonably with the EPA data, as illustrated by the density plot in (Figure 9b). The performance of the local correction model is seen to match the standard correction model, suggesting that over this medium term, relative humidity is probably the primary driver of aerosol changes. In the short-term, the density plot shows that both the standard and local correction models fail to capture the PM$_{2.5}$ distribution accurately. In fact, the use of the correction models then to dampen any contribution of short-term sources to the total signal and increase the difference between the EPA and PA data sets (Figure 9c). This suggests that the primary drivers of short-term fluctuations are particles that are poorly sensed by the PA sensors, and regression-based correction models, including both the standard correction model and local correction models, cannot capture the contribution of those particles.

## 6    Study Limitations

This study has a few limitations. Firstly, the study is limited to one city, and the low-cost air quality sensor network used in the study is not perfectly co-located with the EPA monitoring sites. This can introduce uncertainties in the analysis due to differences in local air properties and pollution sources for the two data sets. Secondly, the placement of the low-cost sensors relative to local built-structures could affect its measurement performance and increase data uncertainty, but this information is not available to us. Thirdly, we did not have access to local traffic-related information or industrial activity, restricting our

ability to strongly relate frequency components to specific emission sources. The likely variability of the local emission sources at the different Purple Air and EPA sites adds uncertainty in quantifying the differences in the short-term responses of the two networks.

## 7    Conclusions

The use of low-cost sensors for air quality monitoring is becoming more widespread and their use has resulted in a better

understanding of air quality at a hyper-local level. Several studies have shown that data from low-cost sensors such as from the purple air (PA) network are less accurate than the gold standard EPA data. Other studies have reported that using correction models, PA data can become comparable to EPA data in accuracy (Mei et al., 2020; Ardon-Dryer et al., 2020; Barkjohn et al., 2021). Understanding the quality of the data reported by low-cost air sensor networks is critical to determining the extent and limitations of the use of this data in policy-making and health studies.

Here, using long-term $PM_{2.5}$ measurements from EPA and PA networks in the Cook County, IL area, we evaluated the accuracy of the reported raw data and recommended correction models. Our initial analysis showed that the corrected PA data was, on average, under-predicting $PM_{2.5}$ by 30 % in the study area. To determine the cause of discrepancy between the PA and EPA datasets, we used a spectral analysis approach to identify the presence of periodic sources i.e,. at 4, 8, 12, and 24 hours in both data sets and then determined their relative response to these sources. Our analysis clearly demonstrates the

PA network's very different sensitivity to different sources. The use of the standard correction model, i.e, US-wide correction model discussed in eq. (1) results in correcting the PA data but significant under-presentation of high frequency sources, particularly traffic. The reason why low-cost sensors may be missing high-frequency components from sources such as traffic can be attributed to several factors. One factor is the minimum detection size limit of the sensors, which is ∼300nm. Sources, such as traffic, with PM emissions predominantly in the sub-300nm size range will, thus, be under-detected in low-cost sensors.

EPA measurements do not have this limitation. Additionally, low-cost sensor response depends on the composition and shape of particles, resulting in PM measurement accuracy varying with emission sources. The implication of these limitations is that the measurements provided by low-cost sensors, such as those in PurpleAir, will be underestimated with respect to certain pollutants, including those associated with traffic emissions, and overestimated related to others. Consequently, relying solely on low-cost sensor measurements without considering the limitations in particle detection and composition could result in an

incomplete understanding of air quality, especially in relation to specific pollutant sources or components.

Also, the standard correction model over corrects for some sources, such as the 12 hour time period source that we identified in this study. Using a local correction model based on temperature and relative humidity, we show that the long-term and medium trends in PA data can be matched with EPA data. In the short-term, both the local and standard correction models perform poorly. The use of both these models actually results in suppression of the contribution of high frequency sources.

Also note, while this study identified several significant peaks i.e., 4, 8, 12, and 24 hours in the power spectrum analysis of air quality data, their precise sources require further analysis and validation.

Our study also demonstrates that, while regression-based correction models maybe seem to improve the accuracy of low-cost sensor network performance by accounting for the contribution of meteorology, they do not uniformly improve the network response to all emission sources. Any field calibration of these sensors using simple regression models cannot correct for this

non-uniform contribution. As best practice, it is recommended that calibration models from field data should report, at a minimum, the distribution of different PM emission sources at that location, and ideally also, the particle size distributions. Given the periodic signatures of many sources, frequency-based scaling approach should be explored towards the development of more robust calibration models that account for the wide range of emission sources common in urban environments. Accuracy of such models will scale with time periods of calibration. Considering the source-dependent response of low-cost sensors,

calibration models developed using land-use data might be an advance over simple regression models.

Thus, care must be taken in using their data in studies where a diversity of emission sources maybe present and their relative strengths are varying over time or space. Advances in sensing technologies and improvements in correction models are critical for expanding our use of data from these emerging low-cost sensor networks.

*Data availability.* The datasets used for this study are available at and can be accessed through the following github repository. `https://github.com/vijaykumar18/Airquality-Spectral-Analysis`. For the entire workflow (reading and organizing data, descriptive analysis, and data analyses) we used the R software (R: A Language and Environment for Statistical Computing) (version 4.2.0), along with the following libraries in our coding: readxl, dplyr, tidyr, ggplot2, car, qqplotr, kza, stats, relaimpo, caret, glmnet,sample, recipes.

*Competing interests.* The authors declare that the research was conducted in the absence of any commercial or financial relationships that could be construed as a potential conflict of interest.

*Acknowledgements.* Vijay Kumar acknowledges the support from US-Pakistan Knowledge Corridor PhD Scholarship Program under Higher Education Commission, Pakistan.

*Author contributions.* VK: Writing original draft, conceptualization, methodology, editing, investigation, analysis, DS: Data curation, visualization, SG: Conceptualization, validation, editing, SS: Supervision, conceptualization, methodology, validation, editing, SM: Supervision, conceptualization, methodology, validation, editing, SD: Writing, review draft, conceptualization, methodology, formal analysis, project administration, All authors contributed to the article and approved the submitted version.

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

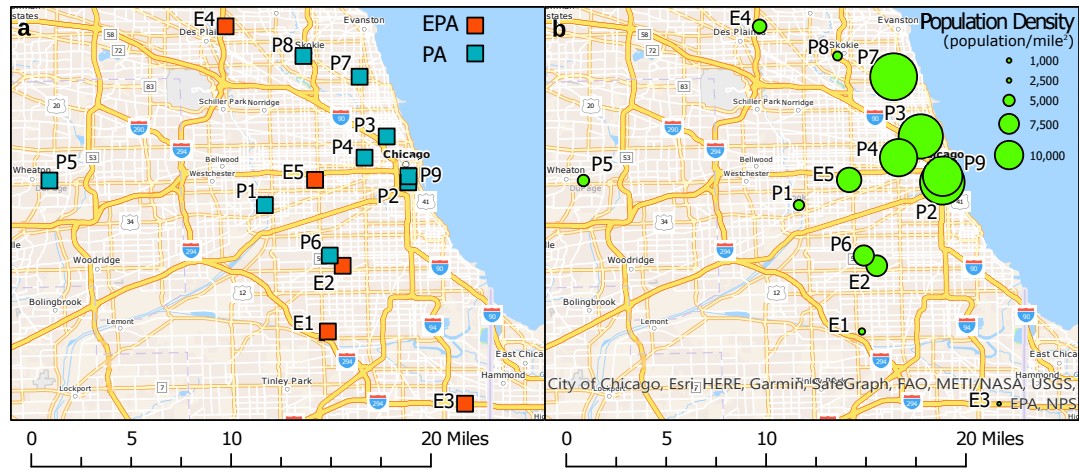

**Figure 1.** (a) EPA and PA sampling locations (b) Population density (population/mile$^2$) in the block defined by U.S. Census Bureau in Cook County, IL. Basemap used from ESRI (ESRI, 2021).

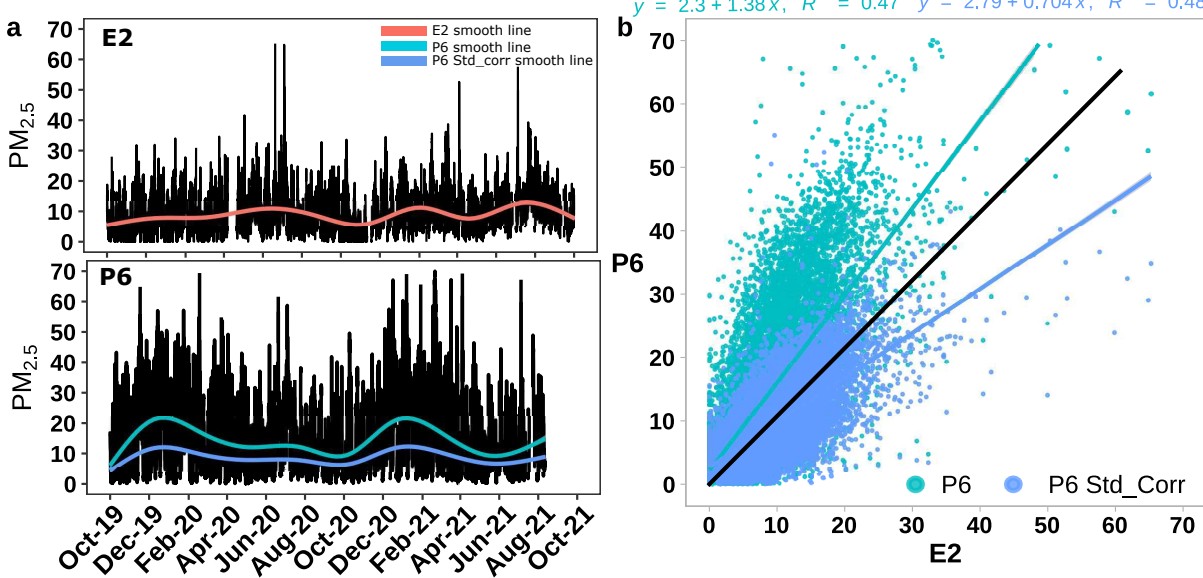

**Figure 2.** (a) Hourly PM$_{2.5}$ measurements from EPA site E2 and PA sensor P6, and (b) hourly PM$_{2.5}$ measurements from EPA site E2 vs PA sensor P6 raw and P6 corrected.

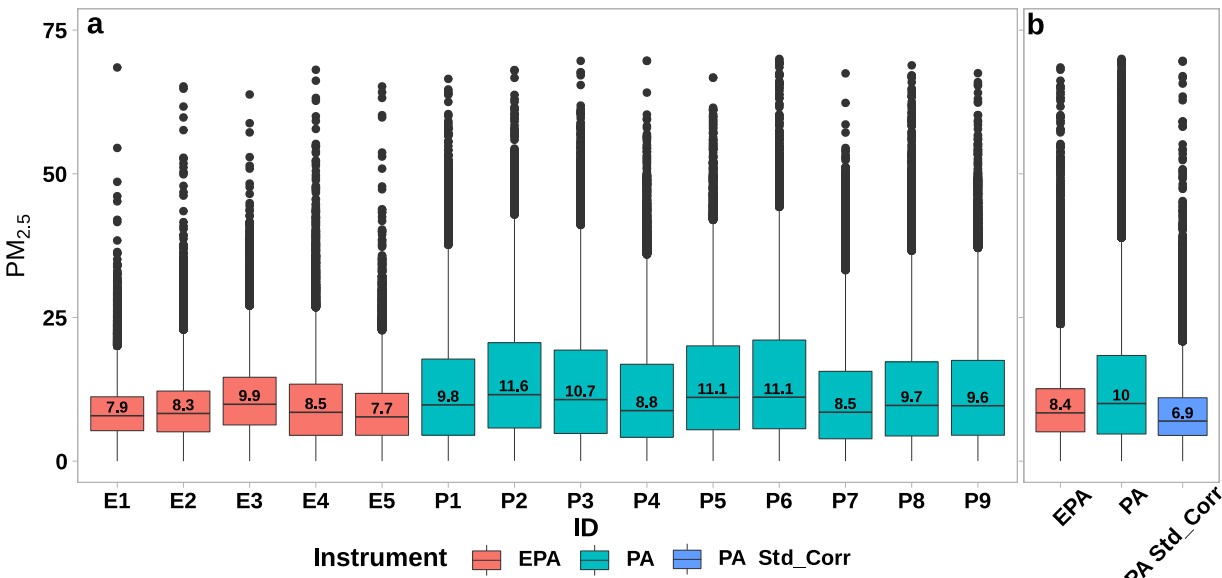

**Figure 3.** (a) Hourly PM$_{2.5}$ measurements from each EPA site and PA sensor located in Cook County, IL (b) all EPA, PA, and PA corrected data together. The box plots represent the overall distribution with quartiles (25th percentile $Q_1$, median 50th percentile $Q_2$, and 75th percentile $Q_3$) values of PM$_{2.5}$ data. The values in black dots over $Q_3$ are outliers.

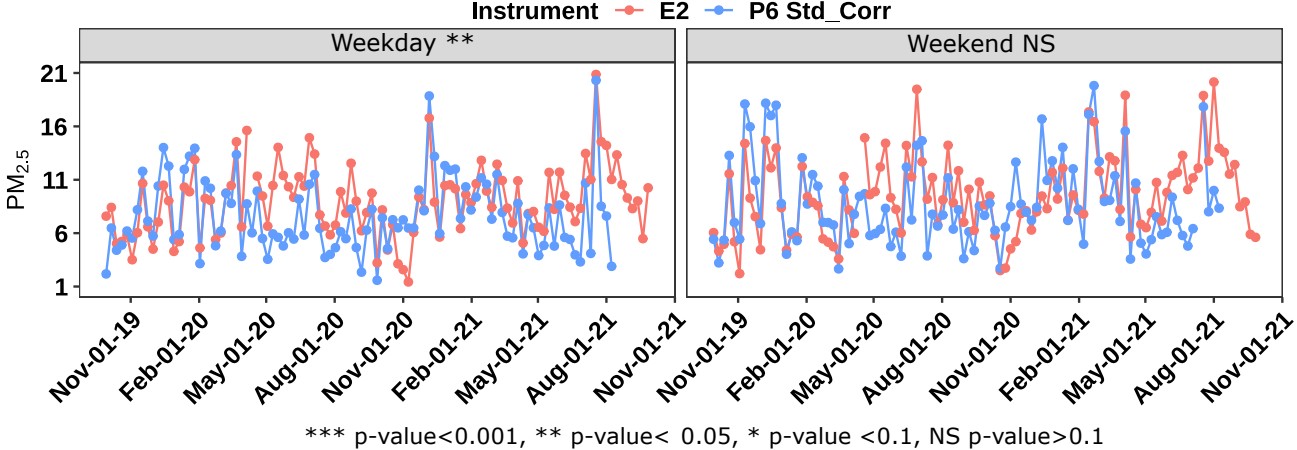

**Figure 4.** Corrected PA sensor $PM_{2.5}$ measurements during weekdays and weekends compared with nearby EPA sites E2, P6. The t-test statistics are provided to determine if there is a statistically significant difference between the two data sets (EPA & PA).

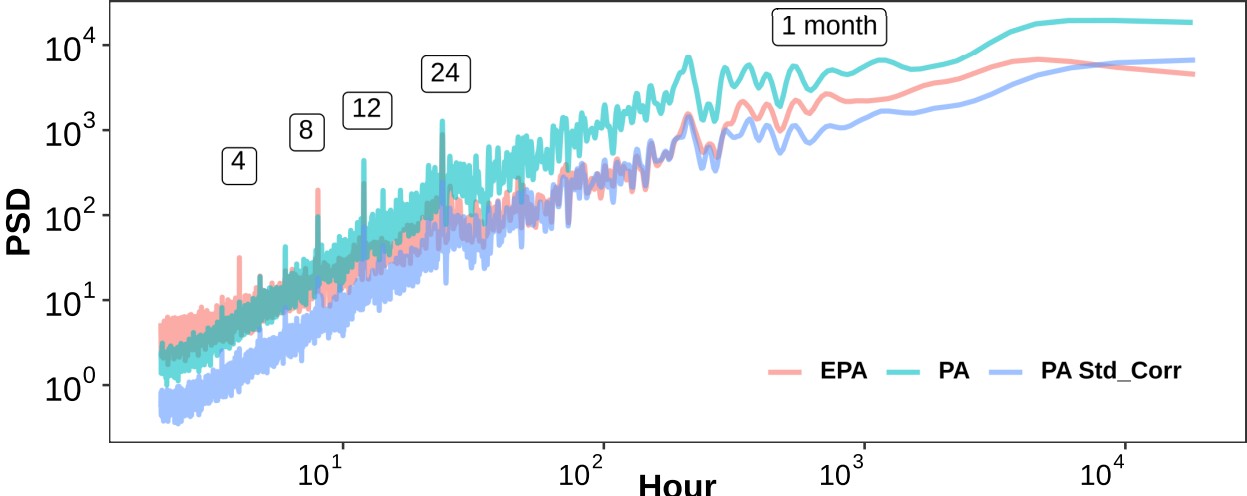

**Figure 5.** Mean PSD of $PM_{2.5}$ data from all EPA sites, all PA sensors, and PA standard corrected data

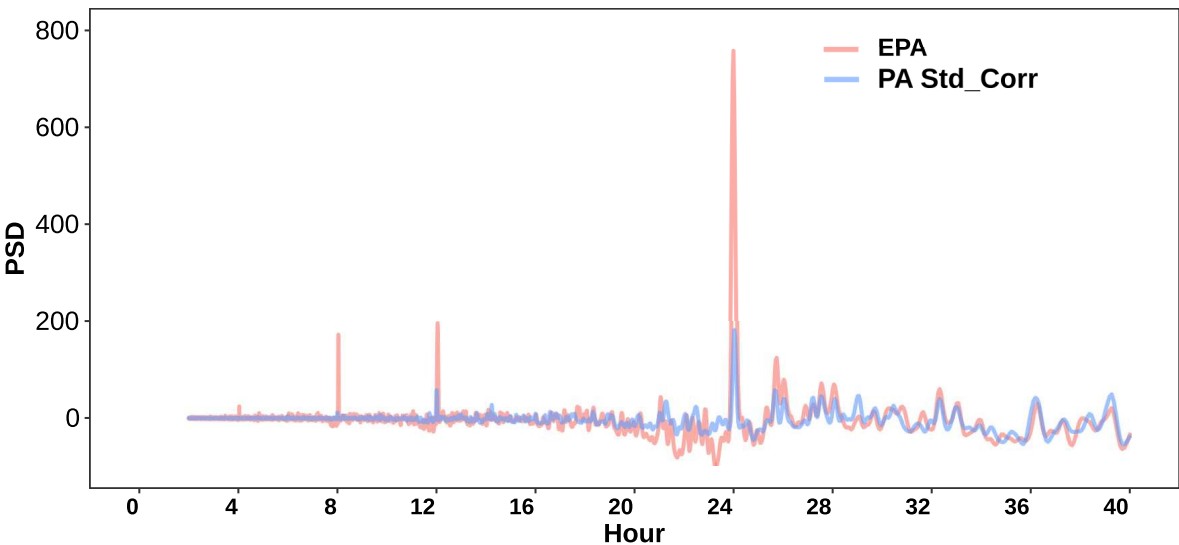

**Figure 6.** Mean PSD of PM$_{2.5}$ data from all EPA sites, all PA sensors, and PA standard corrected data at 4 hours to 40 hours peaks of both networks after removing their baselines.

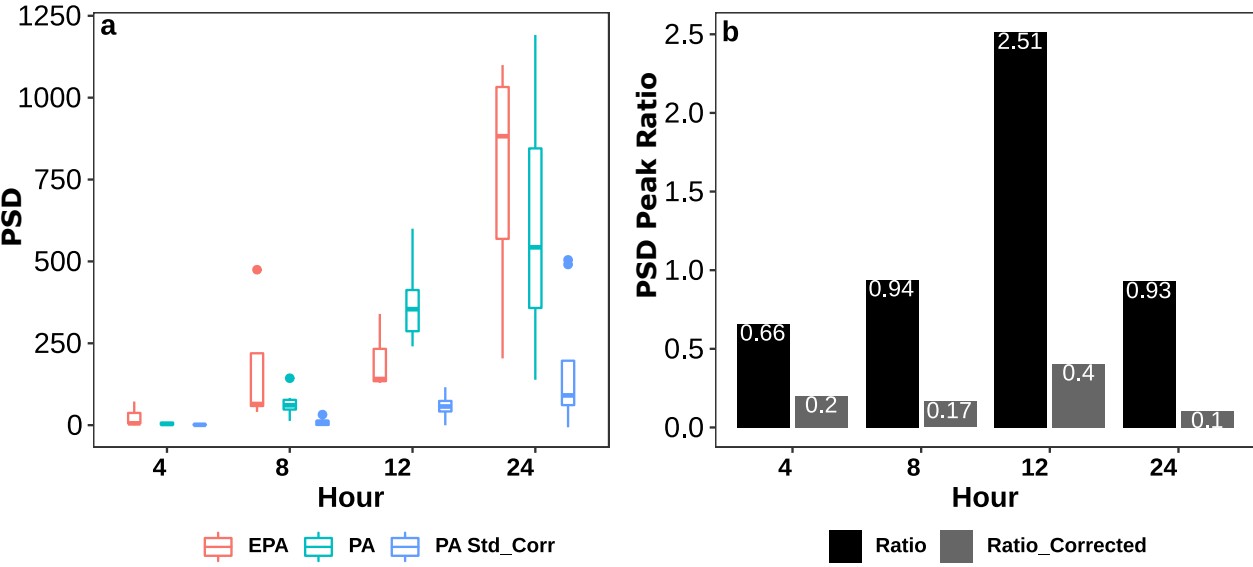

**Figure 7.** (a) Distribution of PM$_{2.5}$ PSD peaks at 4, 8, 12, and 24 hours for all EPA sites and PA locations, before and after correction (b) Ratio of PA to EPA PSD peaks for both raw data (labeled "Ratio") and corrected data (labeled "Ratio_corrected")

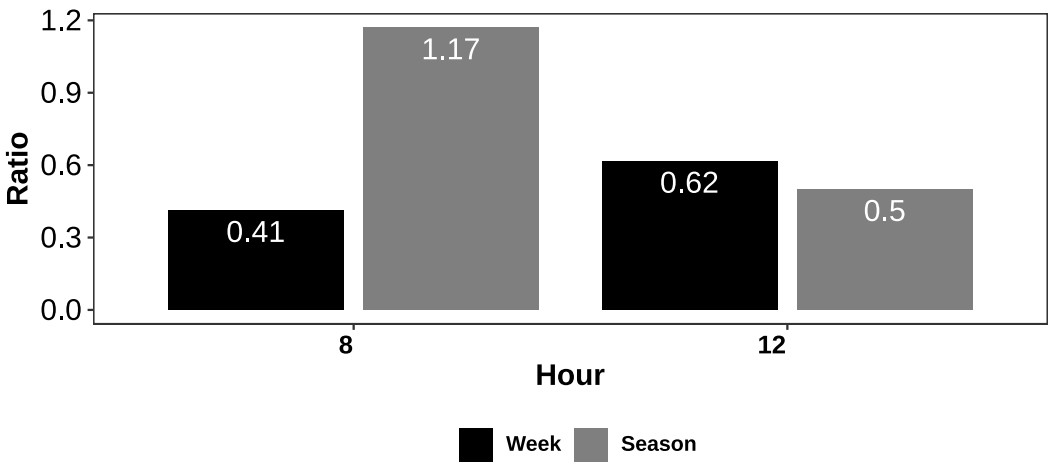

**Figure 8.** Ratio of EPA PSD peaks at 8 hours and 12 hours for the weekend to weekday (labeled "week") and winter to summer (labeled "season").

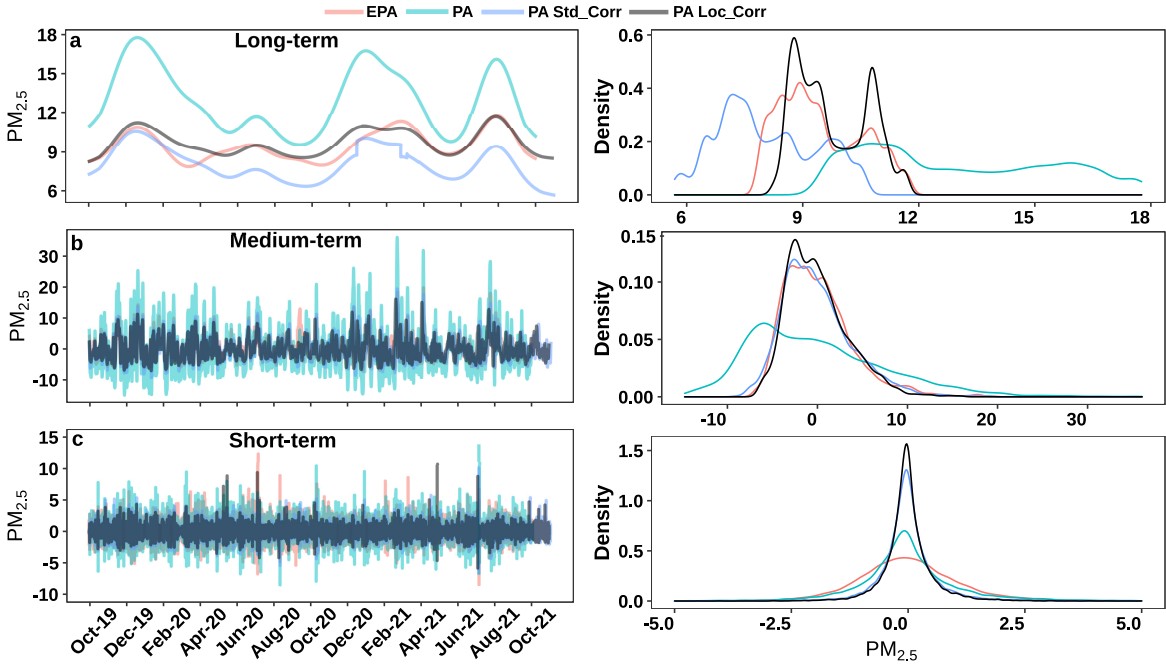

**Figure 9.** Time series and density plot of EPA data, PA data, corrected PA data using standard correction model, and corrected PA data using local correction model for (a) long-term component, (b) medium-term component, and (c) short-term component.