# Peer review of "Spectral Analysis Approach for Assessing Accuracy of a Low-Cost Air Quality Sensor Network Data"

_Atmospheric Measurement Techniques, 2023_

## Author Comment (AC1)

**Response to Anonymous Referee #1**

The manuscript is well-written and clearly structured. Furthermore, the spectral analysis method used to evaluate the data is novel in the context of low-cost sensors. To criticize, the study does not yield scientifically significant new findings. The main conclusion is that the sensor response is source dependent and that without proper calibration, there is a high risk of data misinterpretation. This is the same conclusion that has been made in most, if not all, studies investigating low-cost sensors.

I recommend publication of this study because I consider the approach used to evaluate sensor data valuable. Furthermore, I encourage the authors to consider the following points to strengthen the impact of the research.

We sincerely appreciate the reviewer for dedicating their time to provide detailed feedback. Below, we have addressed each of the specific comments in blue text.

1. The manuscript lacks a clear statement of the limitations of the study. This would be useful for readers to contextualize the findings.

Response: We agree with this suggestion and have included the following paragraph in the manuscript to address the study's limitations.

This study has a few limitations. Firstly, the study is limited to one city, and the low-cost air quality sensor network used in the study is not perfectly co-located with the EPA monitoring sites. This can introduce uncertainties in the analysis due to differences in local air properties and pollution sources for the two data sets. Secondly, the placement of the low-cost sensors relative to local built-structures could affect its measurement performance and increase data uncertainty, but this information is not available to us. Thirdly, we did not have access to local traffic-related information or industrial activity, restricting our ability to strongly relate frequency components to specific emission sources. The likely variability of the local emission sources at the different Purple Air and EPA sites adds uncertainty in quantifying the differences in the short-term responses of the two networks.

2. The authors suggest that the results of their analysis will provide guidance in devising new approaches to calibrate data from low-cost sensors, but it is unclear what specific recommendations are being made . A more explicit discussion of the implications of the study's findings for future research and policy decisions would have strengthened the overall impact of the article.

Response: We have included the following paragraph in the conclusion of the manuscript to discuss the implications of the study's findings for future research and policy decisions.

This study clearly demonstrates that low-cost sensor PM data has non-uniform contribution of different PM sources.  Any field calibration of these sensors using simple regression models cannot correct
for this non-uniform contribution. As best practice, it is recommended that calibration models from field data should report, at a minimum, the distribution of different PM emission sources at that location, and ideally also, the particle size distributions.
Given the periodic signatures of many sources, frequency-based scaling approach should be explored towards the development of more robust calibration models that account for the wide range of emission sources common in urban environments. Accuracy of such models will scale with time periods of calibration. Considering the source-dependent response of low-cost sensors, calibration models developed using land-use data might be an advance over simple regression models.

3.  Line 89 foe;d typo?

Response: We thank reviewer for pointing out this typo, we have fixed it in the manuscript.

4.  1 Consider adding a scale for the map and units for the population density.

Response: We have added the scale and units of population density in figure.

5.  Local correction model; justify the use of both temperature and relative humidity in multiple linear regression. These variables are correlated with each other which can be problematic as the independent variables in MLR should be independent .

Response: We have included the following justification of using relative humidity and temperature in the model to manuscript.

Typically in MLR models, we would only consider independent variables and it could be argued that temperature and relative humidity are not entirely independent. But from a particulate matter perspective, the differing impact of these parameters make them independent of each other. Relative humidity directly affects particle size and hence measurements by low-cost sensors, such as PA. Temperature, however, has a more complex connection to particle properties. Temperature directly affects particle size and composition by modulating condensation/evaporation, which can affect PM measurements by both EPA and low-cost sensors. Temperature also indirectly affects PM properties at a location through its relation to local meteorology, especially wind direction, and hence the distribution of sources at the measurement
location. To establish the independence of these parameters, we calculated the Variance Inflation Factors (VIFs) for temperature and relative humidity and these were found to be below 5. These small VIF values indicate a low level of multicollinearity for the two parameters(Ros-Gálvez, 2017) and permit their inclusion in the MLR model.

6. Line 294 "Our analysis clearly demonstrates for the first time that the PA network's very different sensitivity to different sources." I suggest you remove the "first time" part here.

Response: We have removed the "first time" from our manuscript.

---

## Author Comment (AC2)

**Response to Anonymous Referee #2**

This paper uses spectral analysis to assess the accuracy of low cost air quality sensor networks. This approach is sensible because as the authors note "Sources with short time periods, relative to the calibration period, are averaged out and inadequately accounted for in the calibration. Thus long time scale events are completely lost in the calibration process." The new calibration approach is interesting and appears to have potential.

Response: We sincerely appreciate the reviewer for their valuable time in providing constructive feedback. Our responses to the specific comments are presented below in blue text.

The paper highlights that both the regulatory and low cost networks show distinct peaks in the power spectrum at 4, 8, 12 and 24 hours. It is stated that the 24 hour peak "likely represents harmonics of the 8 hour and 12 hour signals, and hence represents a combination of both (8 and 12 hours) sources". Isn't it more likely that the same sources occur at the same times daily, i.e. morning rush hour, evening rush hour, etc. all start at the same time each day and so you would expect 24 hours to turn up in the power spectrum very strongly?

Response: Yes, we expect the 24 hour peak to be stronger than the 8 and 12 hour peak as is observed in Figure 6. Small differences in the times of rush hour between weekend and weekdays and possibly also within weekdays, could slightly moderate the magnitude of the 24 hour peak.

The 8 hour peak being traffic seems very plausible and the analysis on the days of the week is good evidence due to the preponderance of 8 hour working days. The argument put forward for the 12 hour peak would seem to make more sense to me to be at 24 hour peak, i.e. peak sunlight follows a 24 hour period not a 12 hour period. Could the 12 hour period be representative of 12 hour work shifts, in addition to the 8 hour shifts? Whereas 8 hour shifts represents a standardised "9-5 working day through Mon-Fri", a 12 hour shift is more likely for shift work where workers have a certain number of days on and a certain number of days off, which would not show so significantly in the day of the week effect? The peaks at 4 and 6 hours are also intriguing, what could be causing these?

Response: We agree that the 12 hour peak could also be traffic related. In addition, the 4 and 6 hour peaks also likely have a relation to traffic patterns, as shown in (Lu et at., 2014).

The lack of concrete understanding of the periods should be stated more clearly in the conclusions and abstract. The results are interesting, but far from being definitive as yet.

Response: Yes, we have added the sentence in abstract as well as a paragraph in the conclusions to state lack of concrete understanding of the periods.

With the rationale of calibration periods missing short term signals through the averaging out of signals, it is unclearly why outliers (L100) were removed. These signals could very possible be true, for example due to construction dust etc.

Response: The reviewer raises a good point. Some of the outliers could indeed be real data. Our rationale in removing the outliers was based on following the approach of previous studies, such as the work by (Barkjohn et al., 2021). We believe the major results of our study will not be affected by the treatment of the outliers.

It's not clear why a linear RH correction is used (L118) when the RH effect has been shown to be due particle hygroscopicity, which is non-linear with respect to RH. The k-Kohler approximation well in many low cost sensors, e.g. Crilley et al. 2018 - https://amt.copernicus.org/articles/11/709/2018/ The lack of diurnal RH effect in the correction might explain some of the difference seen in the power spectra of the EPA and low cost data.

Response: It is true that particle hygroscopicity is influenced by RH in a non-linear manner. However, in our study, we intentionally avoided making assumptions about particle properties and their behavior.

The linear RH correction approach was chosen to maintain simplicity and minimize assumptions. This approach has been used in other studies, including the work by (Barkjohn et al., 2021), which we followed for consistency and comparability. By using a linear correction, we aimed to provide a practical and straightforward method for correcting low-cost air quality sensor data without making complex assumptions about particle hygroscopicity.

The authors should provide some rationale why they think the low cost sensors are missing the high frequency outputs. For example, why would the low cost sensors be blind to traffic data for example? Are the low cost sensors not measuring the smallest particles that the EPA sites are measuring? If so, then the low cost measurements are blind to a subset of particles. What is the implication of this?

Response: We have added following paragraph in conclusions.

The reason why low-cost sensors may be missing high-frequency components from sources such as traffic can be attributed to several factors. One factor is the minimum detection size limit of the sensors, which is ~ 300nm. Sources, such as traffic, with PM emissions predominantly in the sub-300nm size range will, thus, be under-detected in low-cost sensors. EPA measurements do not have this limitation. Additionally, low-cost sensor response depends on the composition and shape of particles, resulting in PM measurement accuracy varying with emission sources.

The implication of these limitations is that the measurements provided by low-cost sensors, such as those in PurpleAir, will be underestimated with respect to certain pollutants, including

those associated with traffic emissions, and overestimated related to others. Consequently, relying solely on low-cost sensor measurements without considering the limitations in particle detection and composition could result in an incomplete understanding of air quality, especially in relation to specific pollutant sources or components.

Minor comments

L23-24 the literature around links between air pollution and COVID-19 outcomes are still contested. I would insert a 'maybe' within "…exposure to PM2.5 [might] also impact responses to acute diseases such as COVID-19".

Response: We have fixed this in our manuscript.

L29 – define FRM and FEM and indicate how they differ.

Response: We have added this paragraph to the manuscript.

FRM refers to the specific monitoring methods that have been designated by the EPA as the reference standard for measuring air pollutants, while FEM refers to alternative monitoring methods that have been deemed equivalent to the FRM methods by the EPA. The two methods may utilize different instruments or measurement techniques but have demonstrated comparability in accuracy and reliability. The strict maintenance and calibration routines followed in these stations ensure high-quality data and comparability between different location.

L44 "Using two sensors… allows for the robustness of data collection". Spell out why two sensors improve robustness.

Response: We have added this updated sentences in the manuscript.

Using two sensors that measure the exact same PM measurements allows for the robustness of data collection by minimizing any data noise, loss of data due to sensor failure, or measurement error due to sensor electronics issues.

L80 provide some rationale why Chicago PM2.5 have near doubled from 2017 to 2019.

Response: We have added this into our manuscript.

The likely reason for increase in PM2.5 levels is the associated increase in emissions from mobile sources in recent years (Milando et al., 2016).

L89 typo 'foe;d'

Response: We thank reviewer for pointing out this typo, we have fixed it in the manuscript.

L110 the thermal conditioning within EPA measurements can impact upon particle mass through the removal of volatiles and this should be highlighted.

Response: We have added this information into manuscript

Additionally, temperature and relative humidity can alter particle physical and optical properties that PA measurements are sensitive to. While EPA measurements will also be affected by these air properties, the impact is lower because of thermal and humidity conditioning of samples prior to measurements.

L162 define 'stationarity'.

Response: We have defined the stationarity into manuscript.

Figure 1 – provide units on population density.

Response: We have added the units of population density in figure as well as in the caption.

Figure 2 – define the coloured lines on top of the E2 and P6 time plots.

Response: We have added colored line on top of figure.

---

## Author Response (AR2)

**Response to Anonymous Referee #2**

Overall, I agree with one of the reviewers that perhaps there isn't a significant new finding in the data analysis but that the technique is valuable to have published for others to use. However, I would also say that more clarity should be provided in the text to inform others of how to use this technique/approach. After reading through the paper, I am not sure how the spectral analysis actually assessed the accuracy of the low-cost sensors in a quantitative way; it seems just qualitative at this point. The 4-, 8-, 12-, and 24-hour "sources" are linked to traffic and SOA (as educated guesses, with some literature citations), but there is no summary of what actual error this causes in the PA sensors as compared to the EPA sensor. As I say below, there is a number "17%" that I do not know where it comes from. Figure 9 looks to be the key figure and I do not understand how this gives the conclusions you report. Finally, the conclusions section should be written more precisely and specifically to summarize the specific (and quantitative) points of the paper.

Response: Thank you for the very helpful comments. We have addressed the individual comments below. We agree with the reviewers that this paper is not the first to point out that the low-cost sensors are not accurate or that their measurement accuracy is dependent on particle source. The primary contribution of the paper is the introduction of a new approach to analyze long-term low-cost sensor data that enables quantitative estimation of accuracy of low-cost sensors for sources that have a distinct time signature.

Our responses to the specific comments are presented below in blue text.

Lines 5 and 93 - Is "frequency analysis" and "spectral theory" referring to the same basic thing? Perhaps make the language more consistent in the text.

Response: Yes, both terms refer to the same approach and we agree that consistency is important and we are now using spectral theory in the entire document.

Line 110 - A "cursory" analysis is very unsatisfying here. You can quantify whether or not the PA sensors are in denser populations than EPA sensors. Could this fact alone explain why the PA sensors typically read higher concentrations than the EPA measurements?

Response: We have added quantification to demonstrate that most PA sensors are placed in areas with denser populations compared to EPA sensors. However, the location of sensors in denser populations alone cannot account for the higher concentration, as sensors located in lower population densities also show higher concentrations than EPA sensors.

Line 129 - What is "cf_1"? What is it numerically and conceptually, and is it a constant or does it change in the same way RH does?

Response:  The PA provides two PM2.5 values - one labeled as: cf_1 (higher correction factor) or cf_atm (atmosphere). The two values have different "correction factors" that convert the

sensor light scattering measurements to PM.  For RH less than 70%, both values yield similar results for PM2.5 less than 25 µg/m3.  Outside this range, when the cf_atm and cf_1 start to disagree (Barkjohn et al., 2021). It's important to note that the specific algorithm employed by PA for converting Plantower data into mass concentration, whether using cf_1 or cf_atm correction factors, has not been publicly disclosed (Ouimette et al., 2021)

The standard correction, as developed by (Barkjohn et al., 2021), utilizes cf_1 PA data for the correction equation. Therefore, for the sake of consistent comparison with the standard correction, we have also employed cf_1 for local corrections.

Above text has been added to the updated manuscript.

Line 131 - needs rewording I think - the previous sentence says you use sensors onboard the PA only, but then you use EPA sensors also. I think I know what you did to calibrate the data, but its not clear in the text.

Response:  The correction model only required the use of RH from the PA sensors, not the EPA data.  This is now corrected in the text.

Line 144 - The overestimation by 50% is not at all clear in the figures. Show the fit line equations or state them in the text. Same for the underestimated data.

Response: We have a line with its slope as well as intercept values, the overestimation is around 40% and standard correction has underestimated the actual PM2.5 by 30%. We have updated this in the text.

Line 152 - It is not actually clear to me that the model didn't work. After all, the variability in the data was reduced as compared to the EPA value. The median value is also (very) slightly closer to the EPA value than the uncorrected PA value.

Why did the standard correction model not work? Didn't you use the EPA data to constrain the model and derive a correction factor that minimizes the error between the datasets? By definition then, the model should "work" unless the data were already as close together as they could be. Otherwise, the model is entirely insufficient, which makes me wonder how previous studies could have used this model.

Response:  It must be clarified here that the standard correction model that we used was not built with the EPA data from our study.  We are using this model as a starting point, because this is a commonly used model built with US-wide data sets.  While the model does bring the median values closer, the quality of fit (as represented by $R^2$ value of the correlation before and after correction) does not improve, suggesting that the model does not account for all the causes of differences in the PA and EPA.

Line 160 - Be specific and add the exact p-values to the text and/or the figures

Response: We have added exact p-value to the text.

Line 190 - Does this imply that the model did work? (Line 152)

Response: Yes, as consistently noted in the manuscript, the model does bring the average values of the two networks together.

Also, you can do better than "closer" - be quantitative.
Response: We have calculated the RMSE between EPA PSD and PA PSD, as well as between EPA PSD and PA Standard Corrected PSD. The RMSE between EPA and PA PSD is 237.7, and the RMSE between EPA and Standard Corrected PA PSD is 101.8. This indicates that the PSD of the corrected data is closer to the PSD EPA data. This is now included in the manuscript.

I'm also not even sure this is true; at smaller frequencies, PA looks much better than corrected PA.
Response: That is exactly our observation! The corrections scale down values across all frequencies uniformly, and this is seen to result in a poorer fit at short-term frequencies after correction compared to before correction.

Line 194 - How did you remove the baseline? Is there a single equation you fit?
Response: To create the average PSD curve, we obtained the PSD curves for each of the 5 EPA sites and the 9 PA sites. To remove the baseline of the PSD curve, we first smoothed the PSD data with a moving average filter and then subtracted the smoothed PSD curve from the original PSD curve. For the moving average smoothing, we used a window size of 100 hours.(see supplementary section Baseline removal). We have added this information to the text.

Line 195 - What do the higher peaks mean? This whole section needs a better explanation.

Response: A higher EPA peak indicates that the PSD value in that peak is higher for EPA than for the PA corrected data. We have re-written and re-organized the section to more clearly explain our findings.

Line 201 - Why "Assuming"? Are there rush hours 8 hours apart in the middle of the night also? Where does "17%" come from? (As a number). I also don't understand what "of that in the EPA data" means either.

Response: The reviewer's question is not clear to us. We are assuming that the reviewer is referring to our discussion about the 8 hour peak and wondering if you need 3 peaks separated by 8 hours through out the day to show up as a 8 hour peak in frequency analysis. Assuming that is the question, the answer is "no". Just two peaks with a 8 hour difference in particle

concentration peak (like traffic during the day) will also produce a 8 hour peak in the frequency domain. The paragraph is rewritten for clarity.

It seems the argument is that because there is high traffic, then the PA sensors won't perform well as compared to EPA data. If 17% is referring to the fraction of PM caused by traffic, I would say that doesn't seem like a high value. Also, does that mean you can find a 17% error in the measurements?

Response: The argument is that if the PA and EPA sensors are essentially measuring the same aerosol, the contribution of different sources to their signal should be the same.  Here, they are not. If traffic is assumed to be the only source of the particles with a 8-hour peak in the frequency domain, then the PA sensors are only capturing 17% of the traffic signal.

Where is the discussion of the 4-hr peak? Might all the others be a multiple of whatever is happening here? (I do not see any updates in the text with regards to your response to Reviewer 2 about this.)

Response:  The 4 and 6 hour peaks also likely have a relation to traffic patterns as observed in Sun, (2014) . We have added this to the text.

Why is humidity not a part of the 12-hr peak? RH can profoundly effect low cost sensors and is typically on a 12-hr cycle high to low.

Response: That is a good observation. Humidity could have a 12 hour cycle and could be one of the reasons for the 12 hour peak and this is noted in the updated text.

Line 251 - There really aren't any more details in the Supplementary, just equations. Some narrative description would be nice.

Response: We have incorporated description and some additional information into the supplementary file.

Line 291-292 - this is not a sentence

Response: We have fixed it in the text.

Line 294 - How is drift in sensor performance captured by the local model? Which term of the equation?

Response:  The drift in sensor performance is not captured by any of the models, as we are including elapsed time in the models.

Figure 9 - I do not understand this figure. What is density? Line 298 - I don't understand how Fig 9 proves this.

Response: "Density" represents probability density function (PDF), here that would be the probability of measuring different PM2.5 values during the measurement time period. In Figure 9, the left side shows the time series of the three components of the data while the right side shows the PDF of that data. If the particle concentrations were a constant value throughout the two year time period, then the PDF would be a delta function. The PDFs highlight the variability in the data during the measurement period. Assuming that EPA measurements are perfect, a key finding is that for the short-term time period, after using the country-wide model the PDF of the data is narrower than without correction, suggesting a lot of observed variability is removed by the model (possibly because they were incorrectly attributed to humidity effect). Thus, while the correction model does bring the correlation closer to 1 (Figure 2), it does not equally scale the contribution of all particle sources.

Line 299-300 - this is not a sentence
Response: We have added updated that sentence in the text

Line 320 - Be more specific and precise in the conclusion: which models? Which 12 hour time period source (line 335)?

Response: We have clarified and added more information into text.

Lines 335 and 340 - should not have single sentence paragraphs

Response: We have merged that sentence with the previous paragraph.

Fig 2 - would be helpful to have a 1:1 line displayed

Response: We have added 1:1 line in the Figure 2

Fig 4 - just add actual p-values to the title, remove legend at the bottom (with the ***)

Response: We have included the actual p-values in the text; however, in scientific literature, it is common to express p-values as '***' for easy comparisons. For instance, the p-value for weekdays was 0.0000007535, which can be impractical to write out in full. Rounding it results in '0,' which may not convey the significance properly. Therefore, representing it as p-value < 0.05 provides a clearer indication of its significance.

Fig 5 - is the x-axis frequency?

Response: No, the x-axis is 1/frequency, which corresponds to time in hours.

Fig 6 - All PA data are not shown. The plot goes to 40 hrs, not 24.

Response: That's correct, we have fixed the caption.

Technical corrections
There are still a few formatting corrections that need to be made when the article is typeset, including lines 19, 20, 81, 84, 89, 111, 204, 264

"US" should be "U.S."

Response: We have fixed all mentioned corrections!

Line 175 - you are using PSD before it is defined

Response: Yes, now we have defined in line 177 in updated documents

Line 199 - should be "PA PSD peaks", right?

Response: Yes, that's correct. We have fixed it.